

# Implementation of Polarization Diversity Pulse Pair Technique using airborne W-band radar

Mengistu Wolde[1], Alessandro Battaglia[2,3], Cuong Nguyen[1], Andrew L. Pazmany[4], and Anthony Illingworth[5]

[1]National Research Council Canada
[2]University of Leicester, UK
[3]National Center for Earth Observation, UK
[4]ProSensing Inc., Amherst, Massachusetts
[5]University of Reading, UK

**Correspondence:** Mengistu Wolde
Mengistu.Wolde@nrc-cnrc.gc.ca

**Abstract.** This work describes the implementation of polarization diversity on the National Research Council Canada W-band Doppler radar and presents the first-ever airborne Doppler measurements derived via polarization diversity pulse pair processing. The polarization diversity pulse pair measurements are interleaved with standard pulse pair measurements with staggered pulse repetition frequency; this allows a better understanding of the strengths and drawbacks of polarization diversity,

a methodology that has been recently proposed for wind-focussed Doppler radar space missions. Polarization diversity has the clear advantage of making possible Doppler observations of very fast de-correlating media (as expected when deploying Doppler radars on fast moving satellites) and of widening the Nyquist interval, thus enabling the observation of very high Doppler velocities (up to more than 100 m/s in present work). Cross-talk between the two polarizations, mainly caused by depolarization at backscattering deteriorated the quality of the observations by introducing ghost echoes in the power signals

and by increasing the noise level in the Doppler measurements. In the different cases analyzed during the field campaigns, the regions affected by cross-talk were generally associated with highly depolarized surface returns and depolarization of backscatter from hydrometeors located at short ranges from the aircraft. The variance of the Doppler velocity estimates can be well predicted from theory and were also estimated directly from the observed correlation between the H-polarized and V-polarized successive pulses. The study represents a key milestone towards the implementation of polarization diversity in

Doppler space-borne radars.

## 1 Introduction

The measurement of 3D atmospheric winds in the troposphere and in the boundary layer remains one of the great priorities of the next decade (The Decadal Survey, 2017; Zeng et al., 2016). Such measurements have the potential to shed light on a variety of processes ranging from cloud dynamics and convection to transport of aerosols, pollutants and gases (including

water vapor). Moreover, if assimilated, they can improve the numerical weather prediction of large-scale circulation systems (Illingworth et al., 2018a).





A combination of active systems (radars and lidars) and passive radiometry is currently envisaged to be the best approach in order to provide global observations from satellites. Passive measurements provide atmospheric motion vectors; the technique is well established, well suited to geostationary platforms and benefits significantly from the improved temporal and spatial resolution of current geostationary observing systems [e.g. for the Advanced Himawari Imager on board the Japanese satellite Himawari-8 see Bessho et al., 2016]. However atmospheric motion vectors suffer from height assignment errors which can cause systematic biases (see Illingworth et al., 2018a and references therein).

Active sensors that exploit the Doppler effect and use either aerosol, gas molecules or cloud particles as tracers of the winds have the clear advantage of providing vertical profiles of winds but are more technologically challenging. The ESA Aeolus mission (planned for late 2018, Stoffelen et al., 2005) with its Doppler lidar and the ESA-JAXA EarthCARE mission (planned for early 2020, Illingworth et al., 2015) with its nadir-pointing Doppler W-band radar will offer a first assessment of the potential of such instruments in mapping at least one component of the winds (the line of sight wind in clear air and thin ice clouds for Aelous, the vertical wind in clouds for EarthCARE).

The implementation of Doppler Radar has been a challenging concept to bring to a spaceborne platform (Tanelli et al., 2002; Kollias et al., 2014). This is due to the fast movement of the platform, coupled with the finite beamwidth of the radar antenna, which induces broad Doppler spectra and very short decorrelation times. Due to their high sensitivity and narrow beamwidth for a given antenna size, radars in the W-band frequency will spearhead the implementation of spaceborne Doppler Radar. Despite their ideal properties for spaceborne platforms, W-band radars are still impacted by detrimental effects such as attenuation and multiple scattering (Lhermitte, 1990; Matrosov et al., 2008; Battaglia et al., 2010, 2011). One such implementation is the EarthCARE 94Ghz radar system, where the Doppler velocity will be derived via the standard pulse pair (PP) technique, but it is widely recognized that the same approach cannot be applied to obtain global 3D wind measurements (Battaglia et al., 2013; Battaglia and Kollias, 2014; Illingworth et al., 2018a) nor for the study of microphysical processes (Durden et al., 2016). The radar scientific community has proposed different alternatives to the standard Doppler approach to mitigate issues such as short decorrelation times, non-uniform beam-filling (Tanelli et al., 2002) and aliasing. Two have emerged as the strongest candidates:

1. displaced phase center antennae, which involve the use of two antennae for transmitting and receiving with pulse timing and distance between antennae appropriately chosen to cancel the platform motion effect (Durden et al., 2007);

2. polarization diversity (Pazmany et al., 1999; Kobayashi et al., 2002) with a single (large) antenna.

Polarization diversity (see Fig. 1) exploits the correlation between the backscattering returns from pairs of pulses transmitted with alternating polarization $H-$ (blue) and $V-$ (red), spaced by a short time separation, $T_{hv}$. Because $H-$ and $V-$ polarised pulses backscatter and propagate through the atmosphere independently, the returns from the two closely spaced pulses can be received distinctly by the $H-$ and $V-$receivers. Pairs of $H-$ and $V-$ pulses are transmitted with a low pair repetition frequency (PRF). This practically solves the range ambiguity issues associated with standard pulse pair configurations adopting high pulse repetition frequencies. In fact it decouples the maximum unambiguous range, $r_{max} = cPRI/2$, and the Nyquist velocity, $v_N = \lambda/(4\,T_{hv})$, $c$ being the speed of light and $\lambda$ the radar wavelength. Note that, in order to cancel out phase shifts





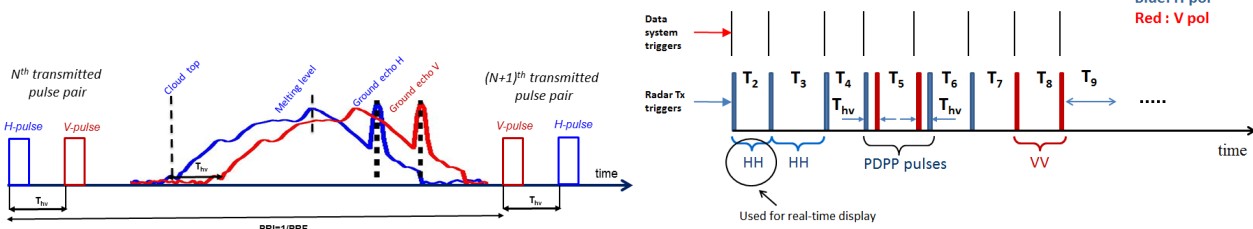

**Figure 1.** Top panel: schematic for the polarization diversity pulse mode. The terms H- and V- refer to the polarization state of the outgoing pulses. Pairs of H and V pulses are transmitted with a pulse-pair interval $T_{hv}$ and with a pair repetition interval $PRI$. Each V-H pair is followed by an H-V pair. Between transmitted pulses, the H (blue) and V (red) receivers sample the backscattered power. Bottom panel: PDPP waveform implemented in the NRC airborne W-band radar system.

occurring in the path between the radar and the scatterers and for any difference in the transmission paths between the two polarizations, $H-$ and $V-$ pairs are interleaved with $V-$ and $H-$ pairs (see Fig. 1).

An airborne demonstration for the displaced phase center antenna concept has been recently completed using Ka-band radar (Tanelli et al., 2016). A ground-based demonstrator for polarization diversity Doppler radar was already available at the
beginning of the millenium at W-band (Bluestein and Pazmany, 2000; Bluestein et al., 2004) and recently the Chilbolton W-band radar have also been upgraded to polarization diversity (Illingworth et al., 2018a). The aim of the current project, funded in the framework of a European Space Agency activity, is to demonstrate the polarization diversity pulse-pair (PDPP) technique using an airborne W-band Doppler radar. Following the successful completion of the ESA PDPP demonstration project, a new satellite concept (WIVERN) - scanning W-band radar operating in PDPP mode has been proposed as a candidate for the ESA
Earth Explorer Mission-10 (Illingworth et al., 2018b).

## 2   Implementation of PDPP on the NRC airborne W-band radar system (NAW)

### 2.1   The NRC airborne W-band radar system

The NRC Airborne W- and X-band Polarimetric Doppler Radar system (NAWX) was developed by the NRC Flight Research Lab in collaboration with ProSening Inc. for the NRC Convair-580 aircraft between 2005 and 2007. A summary of the NAW
system specifications is given in Table 1. The NAWX radar's electronics and data system are rack mounted inside the aircraft cabin while the antenna sub system is housed inside an un-pressurized blister radome (Fig. 2).The NAW antenna subsystem includes three W-band antennae in nadir, aft- and side-looking directions. Two of the antennae, the aft and side antennae, have dual-polarization capability. In addition, a two-axis motorized reflector plate was designed to allow the beam from the aft antenna to be redirected from nadir and up to $50^o$ in the forward direction in either horizontal or vertical planes providing
Doppler measurement at a wide range of incidence angles. The PDPP data were collected using either the dual-polarization



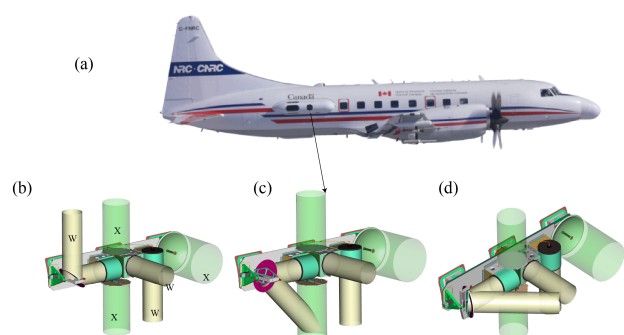

**Figure 2.** The NRC Airborne W- and X-band (NAWX) radar installation inside the starboard blister radome mounted on the Convair 580. The aft antenna beam can be redirected from nadir and up to $50°$ forward along the flight direction.

aft-looking antenna and reflector combinations or the side dual-polarization antenna. Radar beam incidence angles ranging from $0°$ to $80°$ are achieved by performing different aircraft maneuvers (Table 2). Other unique features of the NAW are listed below.

– High quality 1.7 kW peak power air-cooled Extended Interaction Klystron amplifier (EIKA) with a maximum 3% duty
cycle (same as the one used in the CloudSat mission).

– Two channel 12-bit digital receivers with capability of recording radar raw I and Q data for post processing.

– Innovative design incorporating NRC-developed INS-GPS integrated navigation system for accurate Doppler correction.

With its unique capability, the NAW is an ideal platform to demonstrate the PDPP technique for airborne/spaceborne applications. The NAW radar was originally built using a modulator which was not able to double pulse at very short (of the order of
$\mu$s) pulse spacings that is required by the PDPP technique. Therefore, the radar was upgraded with a state-of-the-art modulator allowing the radar to double pulse with pulse spacing as small as $0.5\mu s$. In this mission, PDPP pulse spacing ($T_{hv}$) of 6, 12, 20 and 40 $\mu$s were selected. These specific spacings match integer multiples of the available effective range gate of the radar which is 17.1 m. This eliminates the need to Nyquist sample the return signal and then interpolate the data to co-locate the $V$ and $H-$pol return gates. In order to efficiently evaluate the performance of PDPP technique, a sequence of $H/V$ and $V/H$
polarization diversity pulse-pairs is interleaved with a conventional staggered pulse repetition time (PRT) waveform (bottom panel in Fig. 1). The first three pulses of the waveform form a staggered PRT scheme which extends the unambiguous Doppler velocity range. If the pulse pair processing is applied to a staggered PRT observation, the maximum unambiguous Doppler velocity is determined by the PRT difference (Zrnic and Mahapatra, 1985). Generally, the staggered PRTs are selected as multiples of a curtained unit time. (Zrnic and Mahapatra, 1985) have shown that for the pulse pair technique, the optimal staggered
PRT ratio is 2/3. Therefore, the PDPP waveform was designed such that $T_2/T_3$ is close to 2/3. Additionally, the pulse spacing





**Table 1.** The NRC airborne W-band radar specifications

| | |
|---|---|
| RF output frequency | 94.05 GHz |
| Peak transmit power | 1.7 kW typical |
| Transmit polarization | H or V |
| Maximum Pulse Repetition Rate | 15 kHz |
| Transmitter max. duty cycle | 3% |
| Pulse width | 0.1-1 $\mu$s |
| Antenna ports (electronically selectable) | 5 |
| Receiver channels | 2 |
| Receiver polarization | co and cross-polarization |
| Doppler | Pulse pair and FFT |
| Antennas | 2 x 12" dual-polarization<br>1 x 12" single-polarization |
| Minimum detectable @ 1 km | -30 dBZ (75 m resolution) |

$T_2$ and $T_3$ were set according to the maximum desired measurement range - velocity and the transmitter duty cycle limit of radar.

Pulse spaces should also be small enough to maintain high pulse-to-pulse signal correlation. The normalized signal correlation can be approximated using Eq. 6.5 from (Doviak and Zrnić, 1993) as

$$\rho\left(T_d\right) = exp\left[-\frac{8\pi^2\,\sigma_v^2\,T_d^2}{\lambda^2}\right],\qquad(1)$$

where $T_d$ is pulse spacing (PRI), $\sigma_v$ is the Doppler velocity spectrum width and $\lambda$ is the radar wavelength. At the Convair true air speed ($v_a$) of 100 m/s and antenna beam width $\theta_{3dB}$ of 0.74°, the aircraft motion induced $\sigma_v$ is approximately 0.55 ($\sigma_v \approx v_a\theta_{3dB}/\left[2\sqrt{2\ln 2}\right]$; (assuming a Gaussian antenna beam pattern). In addition, turbulence in the sample volume can further increase $\sigma_v$, so the maximum $T_d$ to maintain greater than 0.9 should be less than about 200 $\mu$s. However the PDPP pulse pairs have to be spaced farther than this, so as not to exceed the maximum average transmitter duty cycle. The selected PRT's for PDPP modes are given in Table 3.

## 2.2 Reflectivity calibration

This section focuses on the calibration of the NAW reflectivity and differential reflectivity measurements. The transmit power, and the gain and noise figure of the two receiver channels were measured by ProSensing Inc. of Amherst, MA, USA using laboratory test equipment, after the PDPP upgrade. Subsequent drifts in the transmitter power were monitored using a coupled detector circuit. Additionally, the receiver gain and noise figure were continuously measured using the Y-factor technique with an internal ambient temperature and heated waveguide terminations in each receiver. The calibration of the remaining




sections, such at the antennae and front-end waveguides, without disassembling the radar, require external reference targets. Water surface and pole-mounted corner reflectors have been tried for calibrations. Using a pole-mounted reflector calibration was problematic due to the difficulty of accurately and consistently pointing a narrow antenna beam at the reflector, while maintaining a high ($>\sim$30 dB) reflector signal to clutter ratio, without saturating the receiver, so this technique was not

employed during the PDPP implementation.

In this work, the end-to-end calibration was done using backscattering properties of the water surface. A detailed description of this method can be found in e.g. (Li et al., 2005; Tanelli et al., 2008). In summary, for 94 GHz radars at a $10°$ incidence angle, the mean value of measured $\sigma_0$ (in clear air condition) is 5.85 dB with a standard deviation of 0.6 dB (Li et al., 2005). Fig. 4 shows the water surface radar cross section ($\sigma_0$) as a function of incidence angle at different polarizations and with respect to

different surface wind directions. It is shown that the horizontal and vertical polarization reflectivity agree very well and there is a crossover point around an incidence angle of $10°$, with near-constant radar cross section regardless of wind direction.

## 3  Field campaign

The airborne PDPP flights were conducted from March 2016 to April 2017. We collected a total of over 31 flight hours of PDPP data (4TB) from 22 flights over diverse weather (clear air, cloud and precipitation systems) and surface conditions (open

water, snow and land). Most of the open water flights were conducted over the Great Lakes region. However, dedicated PDPP flights were also flown over the Pacific and Atlantic Oceans. These flights were conducted in diverse wind conditions, which allowed for characterization of the radar cross section at varying wind conditions. Values of $\sigma_0$ in various surface conditions and sea states at elevation angles ranging from near nadir to 80 degrees were analyzed (Battaglia et al., 2017). In this paper, we focus our observations and analysis of the PDPP data during two weather flights.

As presented in Sect. 2, the PDPP data collection was obtained using two fixed antennae and a reflector that allowed redirecting one of the fixed antennae (aft-looking) into the desired beam angles. Fig. 3 shows a typical flight track during the PDPP data collections, where the aircraft sampled a region of interest by performing a series of horizontal transects, roll sweeps and orbit maneuvers. This allowed accumulating data at various PDPP configurations as well as beam angles ranging from near $0^o$ to about $80^o$. Table 2 summarizes aircraft maneuvers used in the PDPP data collections.

The location and range of the ground return and Doppler velocity from the platform-motion depend on the antenna used and the aircraft maneuver. Fig. 5-6 show examples of the aircraft and ground-beam tracks obtained using the side and aft antennae during a roll sweep maneuver. For the side antenna, the platform motion plus the measured Doppler velocity is generally less than 20 m/s even at the maximum steep roll angle ($\approx$50°) at the Convair's mean true air speed of 100 m/s. In contrast, when using the aft antenna and redirecting the beam forward along the flight direction, the platform motion contribution to the

Doppler velocity can exceed 100 m/s.



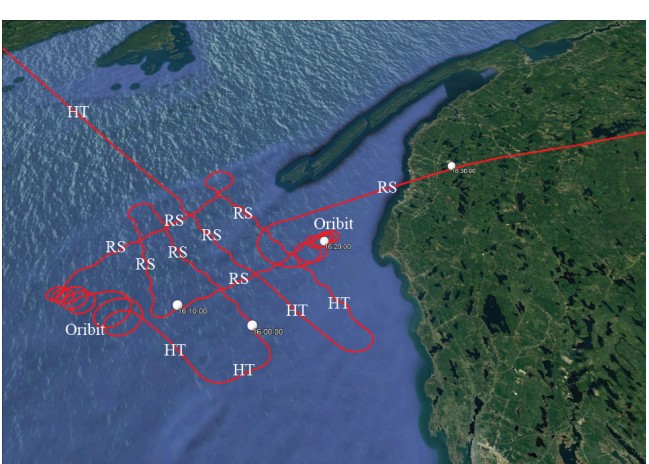

**Figure 3.** Typical PDPP flight track showing where the aircraft performed a series of horizontal transects (HT), roll sweeps (RS) and orbits of various roll angles.



**Table 2.** PDPP operation modes

| Aircraft manoeuver | Antenna | Description | Incidence Angle (°) |
|---|---|---|---|
| Horizontal transects | Aft antenna + reflector | Data collection at different beam angles obtained by changing the reflector position while the aircraft maintains a constant altitude. | 0 to 46 |
| Orbits | Aft + reflector | Aft-looking beam is redirected to nadir view via the reflector. Change in aircraft roll angles from [0-40°]. | 0 to 45 |
| | Side | Change in aircraft roll angles from 0 to 45 degrees. | 45 to 90 |
| Roll sweeps | Aft + reflector | Aft-looking beam is redirected to nadir view via the reflector. Change in aircraft roll angles from [-40°-40°]. | 0 to 40 |

**Table 3.** PRT selection for PDPP modes

| PDPP spacing ($T_{hv}$) [$\mu$s] | $T_2$ [$\mu$s] | $T_3$ [$\mu$s] | $v_N$ PDPP [m/s] | $v_N$ stg. PRT [m/s] | $r_{max}(T_2)$ stg. PRT (km) |
|---|---|---|---|---|---|
| 6 | 90 | 120 | 132.9 | 26.6 | 13.5 |
| 12 | 90 | 120 | 66.5 | 26.6 | 13.5 |
| 20 | 100 | 120 | 40 | 39.8 | 15 |
| 40 | 100 | 120 | 19.9 | 39.8 | 15 |




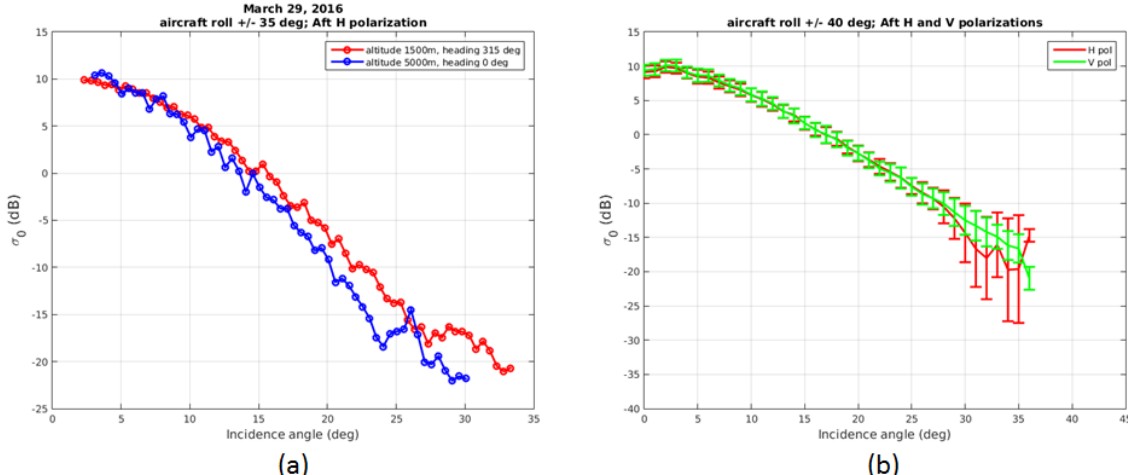

**Figure 4.** Radar cross section of the surface ($\sigma_0$) as a function of incidence angle: (a) measured $\sigma_0$ from H polarization over water surface with different wind direction from March 29, 2017 calibration flight. There is a cross point at incidence angle of around 10 deg and $\sigma_0$ of 5.85 dB. (b) $\sigma_0$ from both horizontal and vertical polarizations for March 04, 2017 flight shows a good polarimetric calibration.



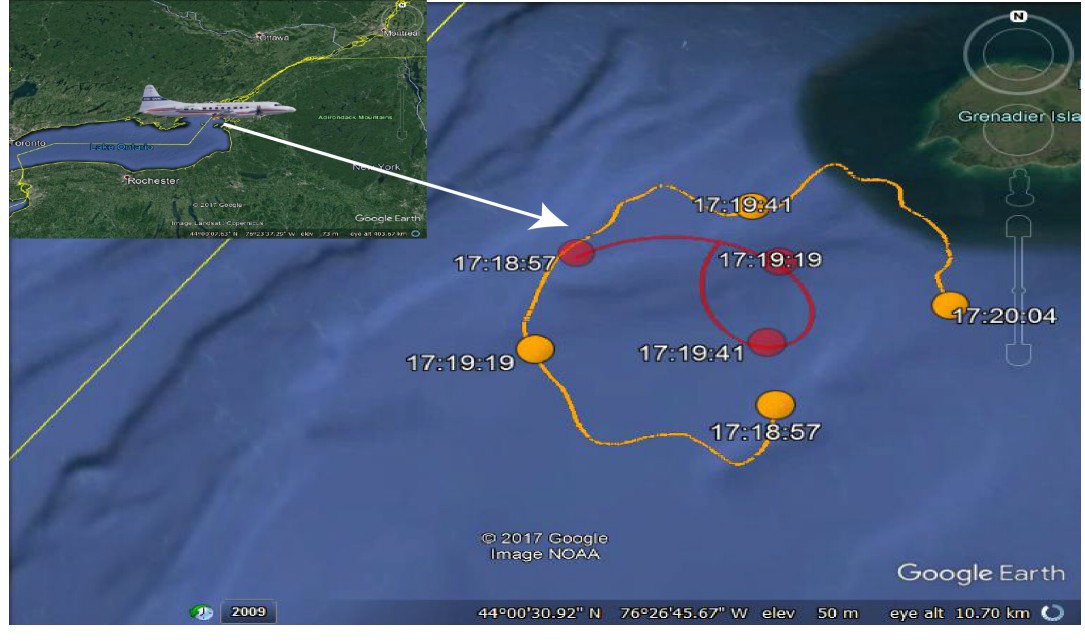

(a)

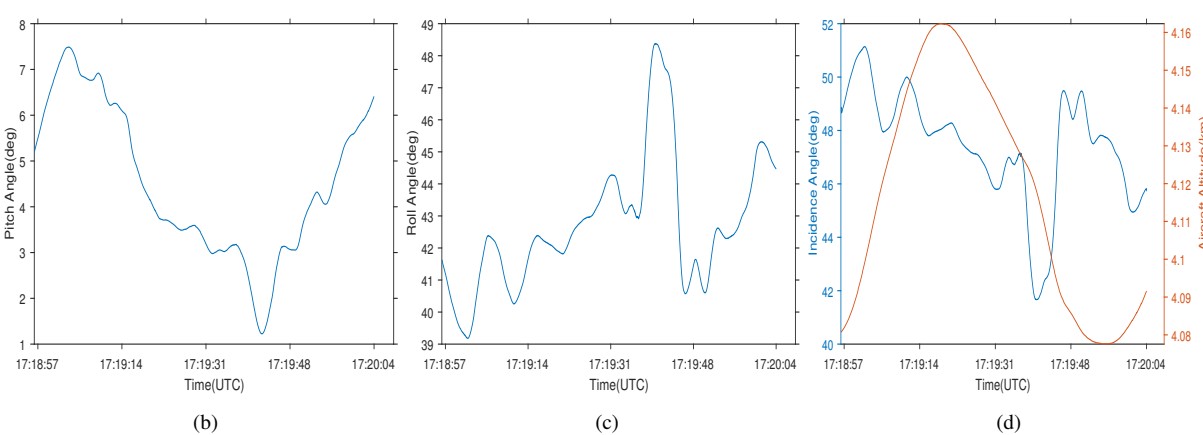

(b)                    (c)                    (d)

**Figure 5.** The NRC Convair-580 flight track (red) and the beam ground track (orange) during a PDPP data collection flight on 10-Jan-2017. The beam ground track show the side antenna beam ground track while the aircraft is performing a partial roll sweep maneuver. The inserts at the bottom of the image show aircraft heading, roll angle, beam incidence angle and aircraft altitude.



(a)

(b)

(c)

(d)

**Figure 6.** Example of beam angle from aft antenna redirected to nadir by the reflector and the aircraft performing a roll sweep from $\sim \pm 45$ degrees over Lake Ontario on 27-Jan-2017.





## 4  Data analysis / Observations

The goals of the field campaign were:

1. to characterize the $\sigma_0$ and the cross pol signatures of ocean and land surfaces;

2. to use the characterization of $\sigma_0$ in order to portray a typical ground surface clutter and the resultant surface blind zone;

3. to investigate the presence of ghosts associated with cross polar returns induced both by the surface and by meteorological targets;

4. to check the validity of the dependence of the variance of the velocity estimates on the signal to noise ratio (e.g. on different reflectivities or different cloud systems), the signal to ghost ratio, the $T_{hv}$ and the number of samples.

The first two goals were addressed in a previous paper (Battaglia et al., 2017); the focii of this paper are on the last two project
goals.

### 4.1  Ghost echoes and impacts on PDPP velocity estimates: theoretical considerations

The key assumption underpinning the polarization diversity methodology is that the $H$- and $V$- pulses are independent. Cross-coupling between the two polarizations can occur either at the hardware level or can be induced during radar beam interactions with the hydrometeors (propagation and/or backscattering in the atmosphere). While the former is typically reduced to values
lower than -25 dB, the latter can be important and is characterized by the linear depolarization ratio ($LDR$). LDR values depend on hydrometeor types and radar beam angles. For example, melting crystals produce high LDR signatures at low to high radar beam incidence angles while for columnar crystals, LDR values increase with radar beam angle. For 94 GHz radars at large incidence angles (e.g. $40°$ for Wivern), atmospheric hydrometeors like melting snowflakes and columnar crystals produce $LDR$ up to -12 dB (Wolde and Vali, 2001a, b). From measurements done by the NRC airborne W-band radar, surfaces tend to
strongly depolarize with characteristic values of -10 dB and -15 dB over land and over sea, respectively (Battaglia et al., 2017). The effect of cross-polarization is the production of an interference signal in both H and V receiver channels. The strength of such interference at sampling time $t$ depends on the strength of the cross-polar signal at the time $t$ shifted by the time separation $T_{hv}$ (forward or backward depending upon whether the receiving channel corresponds to the polarization of the first or of the second pulse of the pair). When converted into range, $r$, the voltages measured in the two orthogonal receiving channels can
be expressed as:

$$\mathcal{V}_H(r) \;=\; \mathcal{V}_{HH}(r) + \mathcal{V}_{HV}(r - \Delta r) + \mathcal{N}_H(r) \tag{2}$$

$$\mathcal{V}_V(r) \;=\; \mathcal{V}_{VV}(r) + \mathcal{V}_{VH}(r + \Delta r) + \mathcal{N}_V(r) \tag{3}$$

where $\Delta r = cT_{hv}/2$ and $\mathcal{V}_{ij}$ is the voltage at the output of the $i-$polarized receiver when $j$ polarization is transmitted, and $\mathcal{N}_V$ and $\mathcal{N}_H$ represent the system noise in the vertical and horizontal receiver channels, respectively. Here we have assumed
that the $H$-pulse is the first pulse emitted (like for the first pair of Fig. 1). The powers can be computed from averaging the





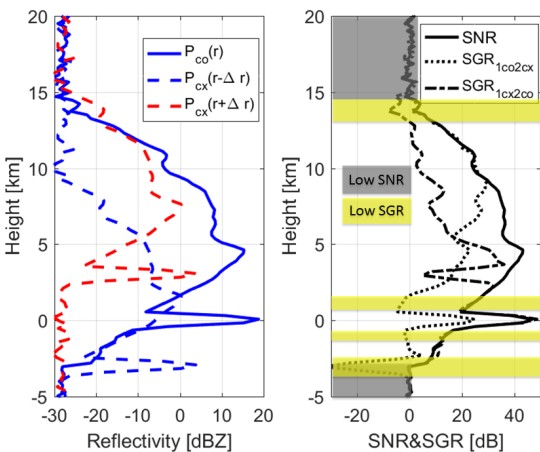

**Figure 7.** Example of cross talk interference in the PDPP scheme for a reflectivity profile extracted from CloudSat (a -28 dBZ equivalent noise power for the $H$- and $V$-channel is assumed). Left panel: the $V-$pulse produces both a co-polar ($P_{VV}$) and a cx-polar return ($P_{HV}$); the same is true for the $H-$pulse. An $LDR$ of -15 dB and a $T_{hv} = 20\mu$ s (corresponding to a height separation of 3 km at nadir incidence) are assumed. Right picture: signal to noise ratio (black) and signal to ghost ratios as defined in Eqs. 4-5. The gray areas identify regions with $SNR < 3$ dB and the yellow bands regions with $SGR$s$< 0$ dB.

module square of the voltages, e.g., $P_V = \langle |\mathcal{V}_V|^2 \rangle$. Under the assumption of reciprocity, no differential reflectivity, and equal gain in the $H$ and $V$-channels, the co-polar power $P_{co}$, and the cross polar power, $P_{cx}$, are polarization independent, i.e. $P_{co} \equiv P_{HH} = \langle |\mathcal{V}_{HH}|^2 \rangle = P_{VV} = \langle |\mathcal{V}_{VV}|^2 \rangle$ and $P_{cx} \equiv P_{HV} = \langle |\mathcal{V}_{HV}|^2 \rangle = P_{VH}$.

In the left panel of Fig. 7 the co-polar power (continuous line) and cross-polar powers (dashed lines) are depicted for a plausible profile (the co-polar profile is extracted from A CloudSat observation) with the cross-polar signals derived under the assumption of a constant $LDR = -15$ dB for the whole profile. There are two possible sources of cross-coupling:

1. The co-polar signal of the $1^{st}$ pulse of the pair (continuous blue) interferes with the cross coupling of the $2^{nd}$ pulse at a range reduced by $\Delta r = cT_{hv}/2$ [dashed blue line, second term on the right hand side of Eq. (2)];

2. The co-polar signal of the $2^{nd}$ pulse of the pair (continuous blue) interferes with the cross coupling of the $1^{st}$ pulse at a range increased by $\Delta r = cT_{hv}/2$ [dashed red line, second term on the right hand side of Eq. (3)].

In the power domain, these interferences contribute to the measured co-polar backscattering signal and sometimes can exceed it (e.g. in the left panel of Fig. 7 near the cloud top at 14.5 km height), thus appearing as "ghost echoes" (Battaglia et al., 2013). To quantify the strength of the interference signals it is useful to define the "signal to ghost ratios" ($SGR$) as:

$$SGR_{1co2cx}(r) = \frac{P_{1co}(r)}{P_{2cx}(r - cT_{hv}/2)} \tag{4}$$

$$SGR_{2co1cx}(r) = \frac{P_{2co}(r)}{P_{1cx}(r + cT_{hv}/2)}. \tag{5}$$





These quantities are depicted in the right panel of Fig. 7 for the profile shown on the left panel.

Because cross-talk signals comes from different range gates, they are independent of the co-polar echoes and do not bias the velocity estimates. However, ghost echoes increase the velocity estimation error as a function of the signal to ghost ratio ($SGR$). There are two possible ways to predict the increase in the variance of the mean velocity estimate. In the first approach,
following (Pazmany et al., 1999), the variance of the mean velocity estimate for the $V-H$ pair, $var_{vh}(\hat{v}_D(r))$, can be estimated as:

$$var_{vh}(\hat{v}_D(r)) = \frac{1}{\pi^2} \underbrace{\left(\frac{\lambda}{4T_{hv}}\right)^2}_{v_N^2} \frac{var(|R_{vh}(r,T_{hv})|)}{2\,|R_{vh}(r,T_{hv})|^2} \tag{6}$$

where $R_{vh}(r,T_{hv})$ is the cross-correlation function at lag $T_{hv}$ which can be estimated as:

$$\hat{R}_{vh}(r,T_{hv}) = \frac{1}{M} \sum_{i=1}^{M} \mathcal{V}_V(r,t_i)\,\mathcal{V}_H^{\star}(r,t_i+T_{hv}) \tag{7}$$

while the variance on the right hand side can be computed as (see Appendix in (Pazmany et al., 1999)):

$$
\begin{aligned}
var(|R_{vh}(r,T_{hv})|) = \\
\frac{P_{HH}(r)P_{VV}(r)}{M}\left[1 - \frac{|R_{vh}(r,T_{hv})|^2}{P_{HH}(r)P_{VV}(r)} + \right. \\
\frac{1}{SGR_{1H2cx}(r)} + \frac{1}{SGR_{2V1cx}(r)} + \frac{1}{SNR_V} + \frac{1}{SNR_H} + \\
\frac{1}{SGR_{1H2cx}(r)\,SGR_{2V1cx}(r)} + \frac{1}{SNR_V\,SNR_H} \\
\left. \frac{1}{SNR_H(r)\,SGR_{2V1cx}(r)} + \frac{1}{SNR_V(r)\,SGR_{1H2cx}(r)}\right]
\end{aligned}
$$

$$\tag{8}$$

$$\tag{9}$$

where $M$ $(V-H)$ pairs have been considered. The expression in Eq. (9) shows that $SGR$ values lower than 1 (0 dB) can become increasingly detrimental for the variance, which is inversely proportional to the number of sampled pairs. A similar expression is valid for the $(H-V)$ pair.

In a second approach, the variance of the velocity estimates is completely characterized by the observed correlation between
pairs of H and V pulses, $\rho_{obs}(T_{hv})$, defined as:

$$\rho_{obs}(T_{hv}) = \frac{|R_{vh}(r,T_{hv})|}{\sqrt{P_H(r)P_V(r)}} \tag{10}$$

which accounts for the effects of the target repositioning (decorrelation), of noise and of the cross-talk introduced by the ghosts. For instance an increase of noise increases the denominator and produces a drop in $\rho_{obs}$. If we can assume independent sample pairs (which is typically true if the $PRI$ is longer than the decorrelation time) then the variance of the pulse-pair velocity
estimates can be expressed as (e.g., Doviak and Zrnić, 1993, Eq. 6.22):

$$var_{vh}(\hat{v}_D(r)) = \left(\frac{\lambda}{4\pi T_{hv}}\right)^2 \frac{1}{2M}\left[\frac{1}{|\rho_{obs}(T_{hv})|^2} - 1\right] \tag{11}$$



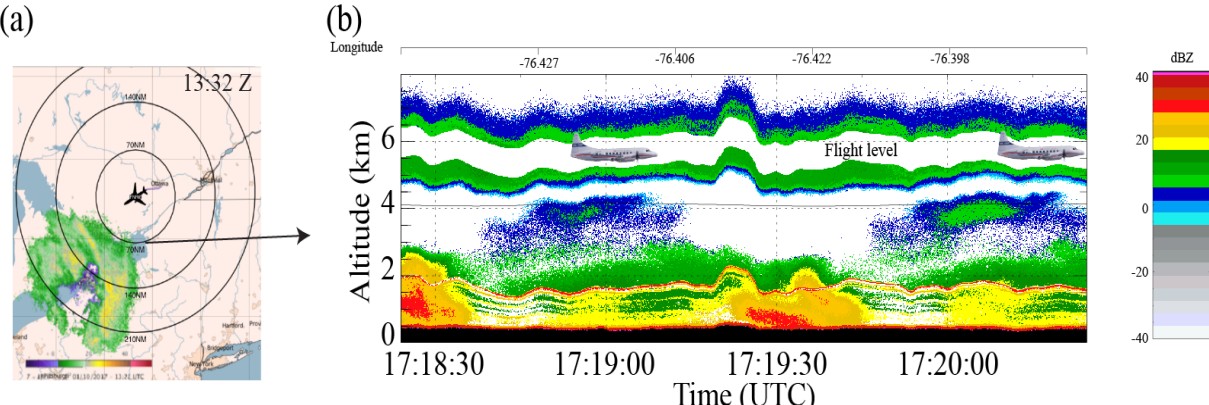

**Figure 8.** (a) Screen capture of NexRad Buffalo S-band Radar received at the aircraft as it leaves Ottawa, ON, Canada on 10-Jan-2017. (b) Vertical cross-section of $Z_e$ obtained by the NRC Airborne X-band (NAX) radar while the aircraft is sampling a cloud system over Lake Ontario.

where $M$ $(V - H)$ pairs have been considered. Since the velocity estimate is obtained from the average of $V - H$ and $H - V$ pulse-pair phase measurements then the variance of the velocity estimate will be:

$$var(\hat{v}_D(r)) = \frac{1}{4} \left[ var_{vh}(\hat{v}_D(r)) + var_{hv}(\hat{v}_D(r)) \right] \tag{12}$$

5    The approaches described above both confer advantages and limitations. Selecting which one to use then depends on the application in question. The first approach is very useful in simulation frameworks; in such conditions the voltage signals described in Eqs. (2-3) can be neatly separated in all their three components and therefore all the terms in Eq. (6-9) can be derived ($|R_{hv}(T_{hv}|$ can be estimated from the Doppler spectral width). On the other hand, the second approach Eq. (11) provides an estimate of the variance of the velocities directly from an observable (the observed correlation, $\rho_{obs}$). This allows an inherent assessment of the measurement-derived Doppler signal quality.

10  **4.2    Field campaign case studies**

In this section, we will present two different flight segments where PDPP data was collected while the aircraft sampled winter clouds.

**4.2.1    Case 1 (10-Jan-2017): Ghost echoes and impacts on PDPP velocity estimates**

In this flight the Convair made extensive samplings of a frontal system over Lake Ontario and its surrounding regions in
15  Ontario, Canada and the state of NY, USA. Fig. 8a shows a screen capture of a Buffalo NexRad image received using the onboard datalink as the aircraft departed Ottawa to sample the precipitation system that covered Lake Ontario and part of Lake





Huron. The NAW was run in PP10 mode for the initial portion of the flight supporting other objectives, and then switched to PDPP mode for the remainder of flight duration, during which the aircraft sampled the tail of a winter storm over Lake Ontario.

During the PDPP data collection, the aircraft first performed repeated roll sweep maneuvers by varying the roll angles $\approx \pm 45°$, and then performed an orbit maneuver using the side antenna. Winds at the flight level of $\approx 5$ km were NW at 30 m/s.

For this case, we selected to highlight the PDPP observations during the orbit maneuver using the side dual-pol antenna. As the W-band measurement was limited to nadir or nadir-fore view, the X-band $Z_e$ profile showing the aircraft altitude with respect to ground and cloud and precipitation structures is shown in Fig. 8b. The aircraft was just below the cloud top, but there was a break in the cloud layer below with a stronger $Z_e$ value extending from the surface to about 1.5 km.

A segment of the flight track performed between 17:19 and 17:20:04 UTC over the northeastern bank of Lake Ontario is

shown in Fig. 5. For this data file, $T_{hv}$ is set to 12 $\mu s$ (equivalent to 900m). The aircraft performed a complete circle with roll angles between 39° and 49°, at an altitude of 4 km. As a result, the range to the surface is changing between 5.8 and 6.2 km.

The received powers in the H and V-channel when the radar was operating in PDPP mode are shown in Fig. 9. Even if the roll/beam angle changes are modest during the aircraft orbit maneuver, there is still significant change to the surface $Z_e$ values, likely due to a change in surface water wave patterns, water surface targets such as boats, and other ground targets.

The linear depolarization ratio (not shown) clearly shows enhanced areas of $LDR$ (approximately between -15 dB and -12 dB) in correspondence to the surface. In contrast, $LDR$ from hydrometeors were not detectable except when at close range to the aircraft Note that the H-receiver must be turned off when the V-pulse is sent out, which corresponds to a blind layer in the left panel. Since the power $P$ is depicted (and not reflectivity), targets at close ranges produce larger signals (since $P \propto \dfrac{1}{r^2}$).

The ghost echo/returns shown in Fig. 9 appear in both $H$ & $V$ channels at different ranges. As explained in Sect. 4.1 these

ghosts are the result of cross talk occurring at the same range increased (left panel) or decreased (right panel) by $cT_{hv}/2$ which in this case is equal to 1.8 km. Two kind of ghosts are clearly detectable (see white arrows): those related to the surface and those produced just after a range equal to $cT_{hv}/2$ associated with the enhancement of backscattered power by targets at very short ranges. The latter are spurious effects which will not be produced in a space-borne configuration (when all targets are basically at the same distance).

The signal to ghost ratios, as defined by Eqs. (4-5), are plotted in two panels of Fig. 10. The areas with blue colours correspond to regions where the magnitude of the ghosts is comparable or larger than the magnitude of the signal and will be therefore characterized by a significant reduction in Doppler accuracy, as previously discussed. On the left panel the extended area of small $SGR$ just after a range of 1.8 km is caused by the interference of the cross-polar signal at very short ranges; other ghosts are present below the surface range. On the other hand, the key features observable on the right panel of Fig. 10 are

found in correspondence to ranges 1.8km shorter than the surface range. This is caused by the strong cross-pol signal of the first pulse in the pair.




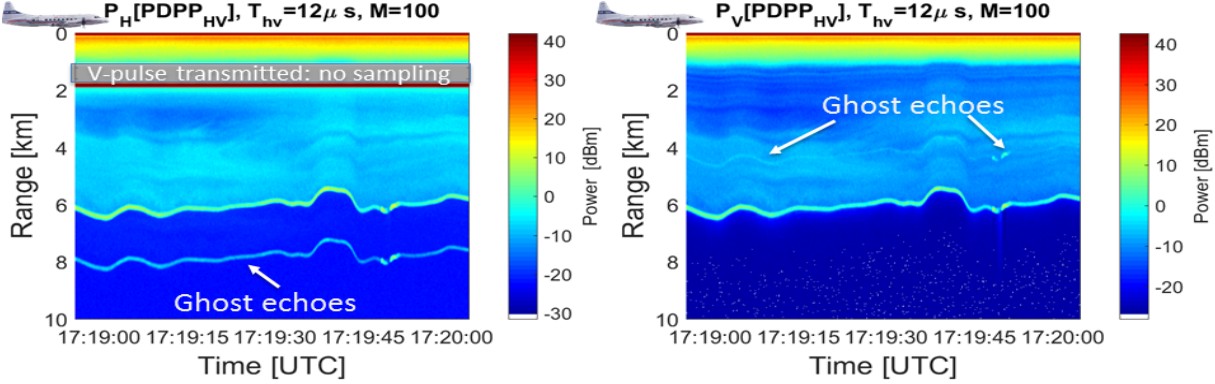

**Figure 9.** Example of power as recorded when operating in PDPP mode with H-pulse followed by an V-pulse after 12 $\mu$s. Left panel: power received in the H-channel; right panel: power received in the V-channel. The mechanism for producing ghost echoes is explained in the upper part of the panel.

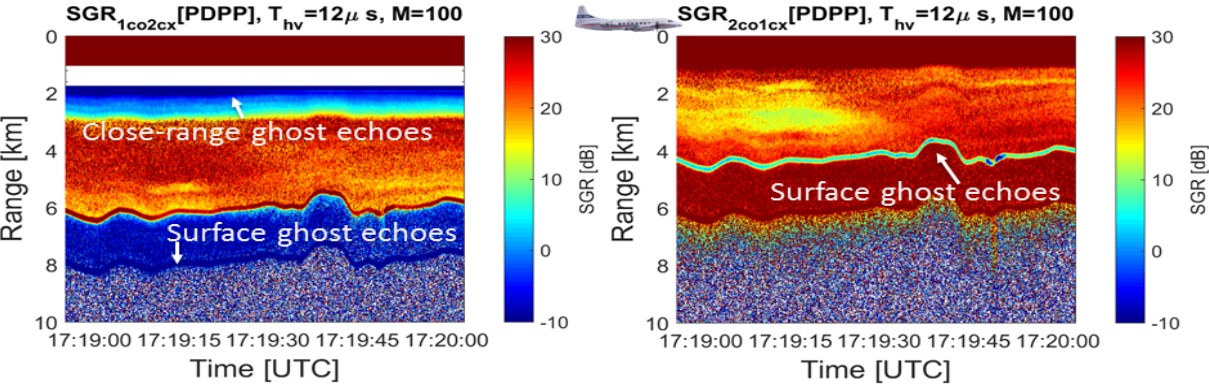

**Figure 10.** Signal to ghost ratios as defined by Eqs. (4-5). Areas with blue colours will have a significant reduction in Doppler accuracy due to the additional noise produced by the ghosts. Note on the right panel the extended area of small SGR caused by the interference of cross-polar signal at very short ranges.





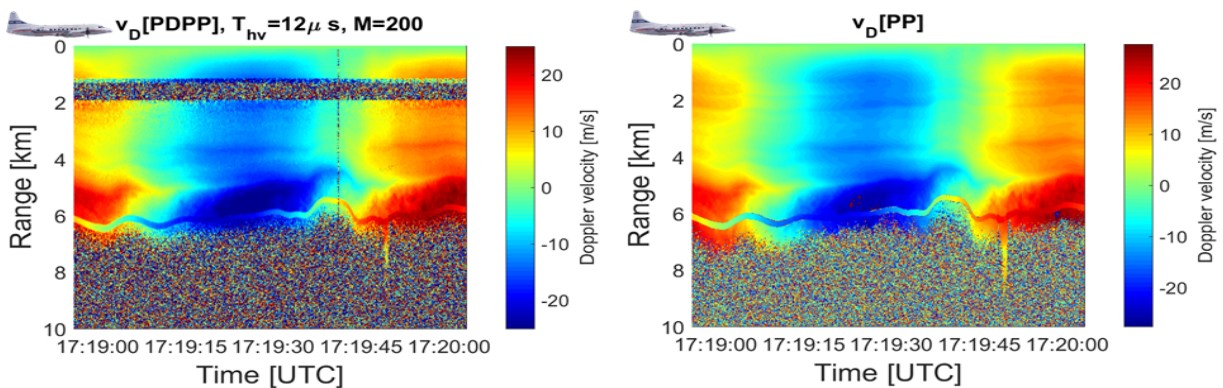

**Figure 11.** Velocity as measured by the PDPP technique with $T_{hv} = 12 \ \mu s$ (left) and by the pulse-pair with two staggered pulse repetition times of 90 and 120 $\mu$s (right).





Fig. 11 shows a comparison of the Doppler velocities as measured via a PDPP sequence with $T_{hv} = 12~\mu s$ ($v_N = \pm 66.5$ m/s) and a conventional PP technique using two staggered $PRF$s($\pm v_N = 26.6$ m/s) (see (Torres et al., 2004) for details). The line of sight winds are seen to change as expected, with a sinusoidal pattern in time, clearly mirroring the change in the heading of the aircraft (see Fig. 5). The estimates have been done using 100 H-V plus 100 V-H pairs and 100+100 staggered conventional

H/H and V/V pairs (corresponding to roughly 20 km integration in the Wivern configuration). The $V_d$ from PDPP is similar to the one estimated using standard $PP$, except for those ranges when the receiver was turned off because of the transmission in the other channel. In that situation the velocity is a random number within the Nyquist interval (like in all the other regions dominated by noise).

For better comparison we have used the same scale between -26.6 and 26.6 m/s for the Doppler velocities (but in the left

panel the velocities range between -66.5 and +66.5 m/s). In this scenario there were no high winds and therefore the staggered pulse pair Nyquist interval was good enough for measuring the Doppler velocities (though there are few aliased points close to the surface at about 17:19:22 UTC), but of course the PDPP has the potential to unambiguously measure much higher velocities. However, this improvement is not without drawback, as PDPP produces a noisier estimate of the Doppler velocities (compare the two panels), particularly in presence of low $SGR$s (e.g. $SGR < 0$ like in correspondence with the blue-coloured

regions highlighted by the white arrows in the two panels of Fig. 10).

As a proxy for the velocity estimate standard deviation, the standard deviation was computed for each point, using a $3 \times 3$ pixel window centered on each position (in time and space); results are shown in Fig. 12 for the PDPP (top left) and the $PP$ (top right) estimates. This will be referred to as the velocity standard deviation estimated from the spatial/temporal variability. This method might overestimate the velocity standard deviation, because it has enhanced values in correspondence to strong

spatial gradients of the Doppler velocities (e.g. in the region about 1 km above the surface). The $PP$ is clearly significantly better (note that the colorbar scale is ranging from 0 to 1 m/s), with the PDPP performing particularly poorly in correspondence to the surface and to the close range ghost echoes. On the other hand, the velocity standard deviation can be computed using either Eq. (9) (see bottom left panel) or Eq. (11) (see bottom right panel). In the first case all the quantities that appear on the right hand side of Eq. (9), except for $|R_{vh}(r, T_{hv})|$ which is estimated via Eq. (7), must be computed from the measurements

in $PP$ mode. This requires firstly an estimation on a ray by ray basis, for the noise levels in the two receiver channels, and secondly an estimate of the noise-subtracted co- and cross-polar powers. Finally, from these estimations, $SNR$s and the $SGR$s are both computed. Both techniques produce results very similar to the velocity standard deviation estimated from the spatial variability with the largest discrepancies concentrated in areas where the Doppler velocity field is rapidly changing (compare with the left panel in Fig. 11), though the results in the left bottom panel are noisier. Again, it is useful to underline that Eq.

11 can be directly applied to an observed variable (the observed correlation); not only does the PDPP allow estimates of the Doppler velocity but also its expected accuracy. Again it is useful to underline that Eq. (11) can be directly applied to an observed variable (the observed correlation); not only the PDPP allows to estimate the Doppler velocity but also its expected accuracy.



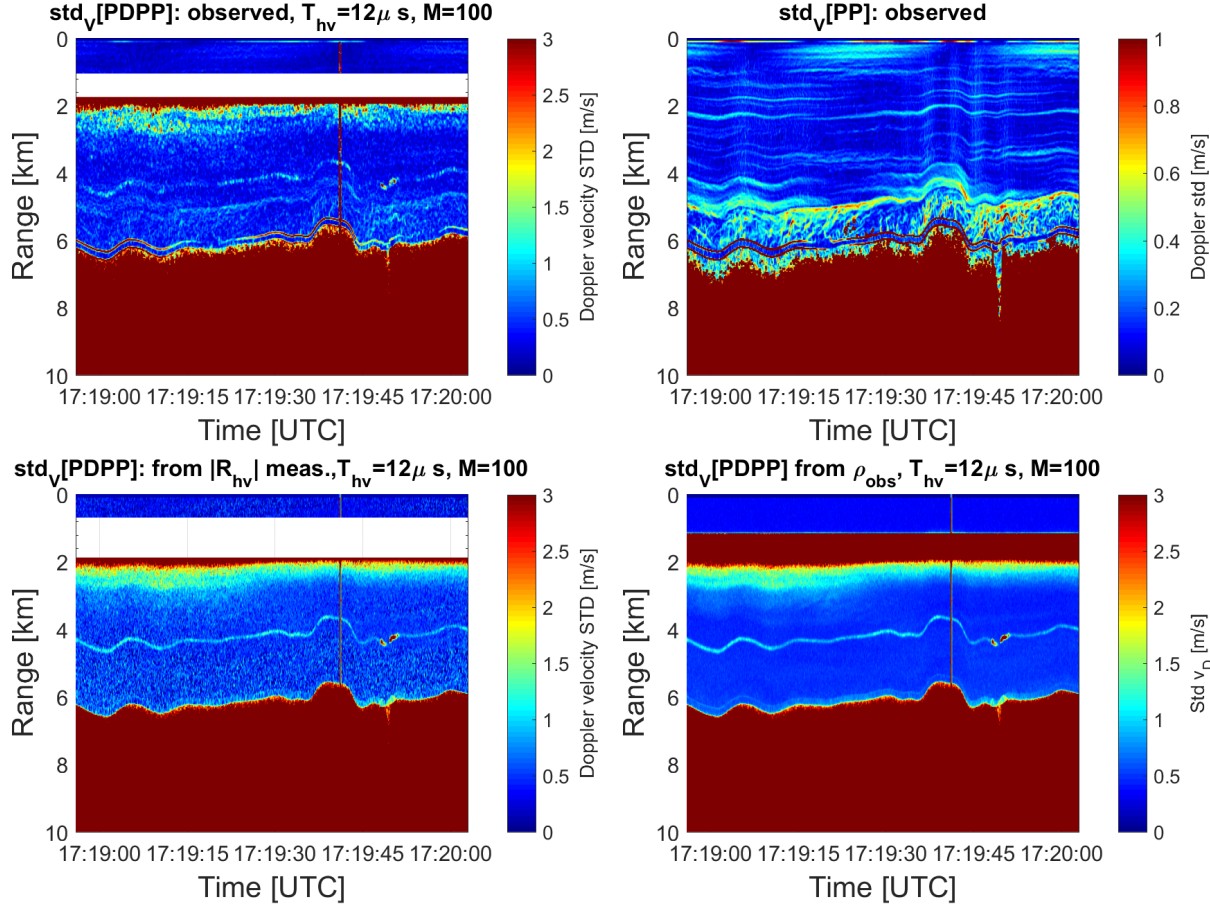

**Figure 12.** Top panels: standard deviation of Doppler velocity estimated from the spatial variability of the velocity fields shown in Fig. 11 for the PDPP (left) and $PP$ (right) technique, respectively (note different scales for the colorbar). Bottom panels: standard deviation of Doppler velocity as derived from Eq. (9) (left) and Eq. (11) (right).

### 4.2.2 Case 2 (25-March-2017): extreme Doppler velocity measurements

For this case, we highlight instances of high $V_D$ from the PDPP measurement inside a major winter storm on 25 March 2017. The Convair flew for over 5 hours sampling the winter storm over land near Buffalo, NY, over Lake Huron and Lake Ontario as the frontal system moved towards south-eastern Ontario. Fig. 13 shows a screen capture of the US Buffalo NexRad $Z_e$ and the $Z_e$ profiles of the NRC Airborne X-band (NAX) radar corresponding to the PDPP data segment of Case 2. The NAX $Z_e$ imagery shows the aircraft descended from about an altitude of 4.8 km to 4 km and remained in cloud above a well defined melting layer ( 2.5 km.). The PDPP data was collected using the Aft antenna and reflector combinations while the aircraft is performing a horizontal transect and descending from 5 to 4.4 km.



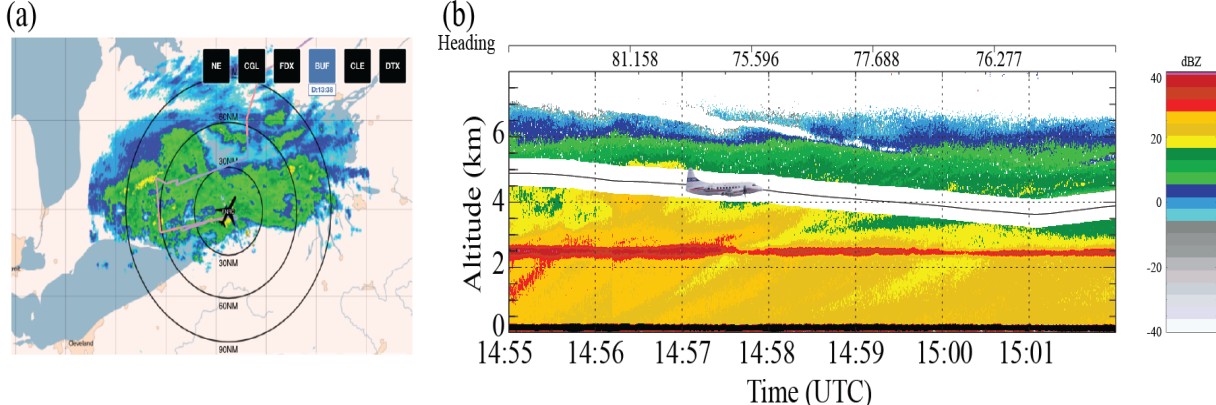

**Figure 13.** Same as Fig. 8 except during the PDPP data collection inside a winter storm on 25-Mar-2017.

The use of the aft antenna in combination with the reflector provides a unique means to direct the antenna beam to a wide range of positions. By steering the aft antenna's beam to a large slant angle, the maximum Doppler velocity introduced by the aircraft motion can reach up to 100 m/s, which makes a good test for evaluating the PDPP method's high velocity retrieval capability. Fig. 14 shows the aircraft flight track, the beam ground intersection track, the aircraft altitude, as well the beam

incidence angle during the March 25, 2017 flight. The beam position began at nadir, and was steered to the reflector limit(50° incidence angle) at which point the beam was then moved back to the nadir position in step decrements of 10°. The PDPP spacing ($T_{hv}$) for this case was set at 6 $\mu$s(900m) which provides a maximum unambiguous Doppler of 132.9 m/s (Table 3).

Fig. 13 shows PDPP reflectivity fields measured during this stepped change in the reflector's angle. There was a well-defined melting layer at around 2.5 km (consistent with the NAX $Z_e$ profile shown in Fig. 13b), which can be seen at beginning of this

segment when the antenna beam was at nadir position. As the antenna beam moved forward toward the flight path, the ground and melting layers appear at different radar range. Similar to the case study in Sect. 4.2.1, ghost echoes associated with the ground and melting layer are observed in the second PDPP pulse reflectivity ($Z_{hh}$ and $Z_{vh}$ in Fig. 15).

The measured PDPP Doppler velocity was properly retrieved using a method described in (Nguyen and Wolde, 2018) is depicted in Fig. 16a. The measured Doppler was as high as 100 m/s in regions where the reflector was steered at $\approx 50°$ and

reduced to a few m/s when the reflector moved back to nadir position. In all scenarios, PDPP technique works very well. There was almost no velocity folding even at weak signal regions (around 14:57:12 UTC and at 6 km altitude). A comparison of $V_d$ obtained from PDPP with the conventional staggered PRT techniques is shown in Fig. 16. Due to a much narrower Nyquist range, staggered PRT velocity was folded many times. This case is a successful example of using the PDPP technique to measure very high radial Doppler velocity in order to obtain accurate horizontal winds from space.

Fig. 16b depicts the Doppler velocity after removing the aircraft motion. Once the aircraft contribution into the Doppler velocity is removed, the $v_D$ estimates are a combination of the hydrometeor's terminal velocity as well as the wind along the line of sight. In this case, the hydrometeor's terminal velocity can be neglected except in rain, below the melting layer, and




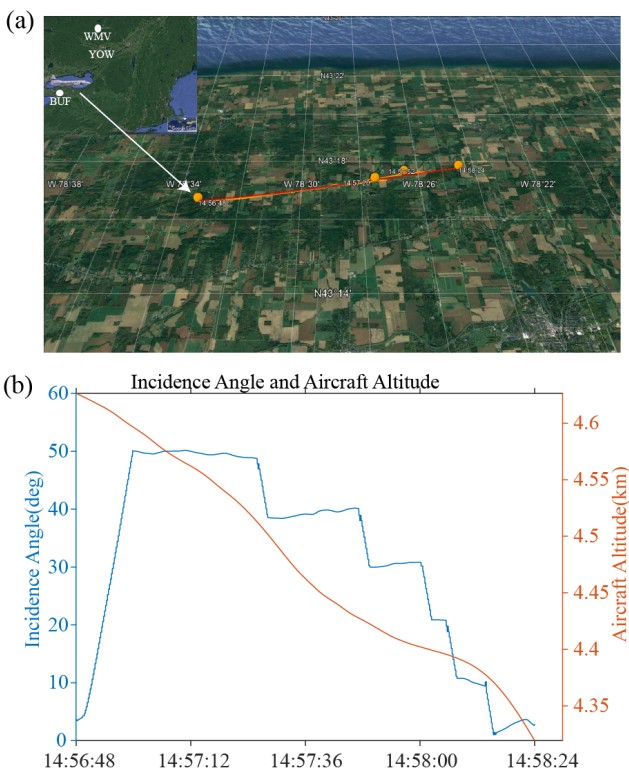

**Figure 14.** Flight track and beam configuration during the PDPP data collection on 25-Mar-2017 (a) The NRC Convair-580 flight track (red) and the radar beam ground-track (orange). The locations of the two radiosonde stations, Buffalo, NY (BUF) and Maniwaki, Quebec (WMV) and Ottawa (YOW) are also shown to aid the interpretation of the data presented in Fig. 13. (b) Aft-antenna beam incidence angle and aircraft altitude.

when the beam is in nadir position. It can be seen that the strongest Doppler velocity after removing the platform motion was recorded when the Aft antenna beam was redirected nearly 50° forward along the flight direction (14:57:00 - 14:57:24). In this high $v_D$ segment, the $v_D$ was negative (away from the radar) between flight level with slightly decreasing magnitude from -20 m/s at the flight level to nearly 0 m/s at around 5.5 km range. The $v_D$ became positive (towards the radar) between the surface 5.5 km radar range. The PDPP $v_D$ profiles are consistent with the vertical profile of horizontal wind measured by the aircraft and radiosonde soundings of nearby stations (not shown).





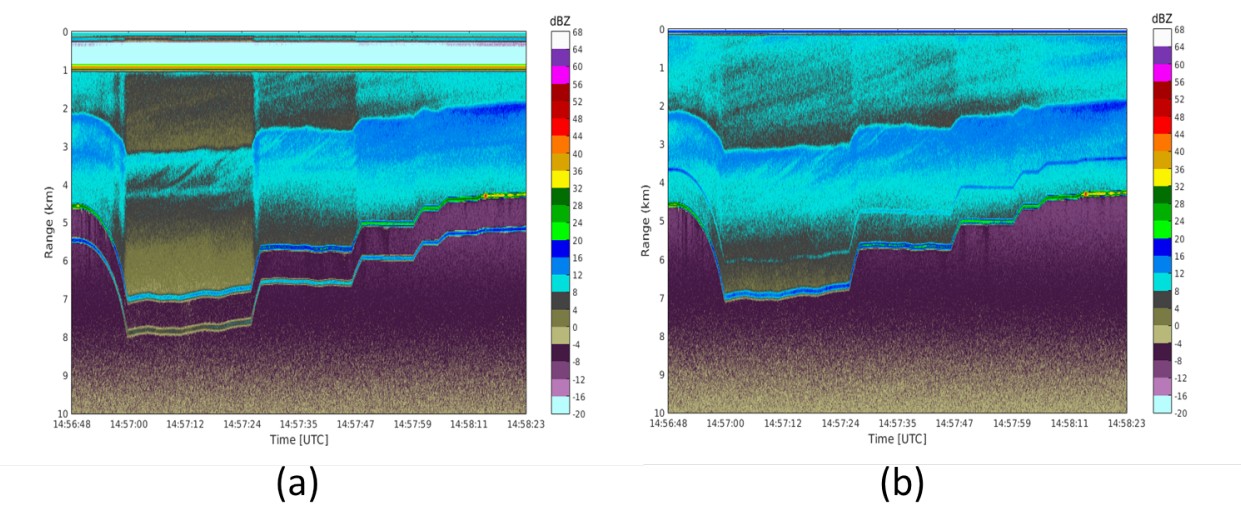

**Figure 15.** Similar to Fig. 9 but for March 25 flight and in reflectivity space. The PDPP spacing in this case is 6 $\mu$s.

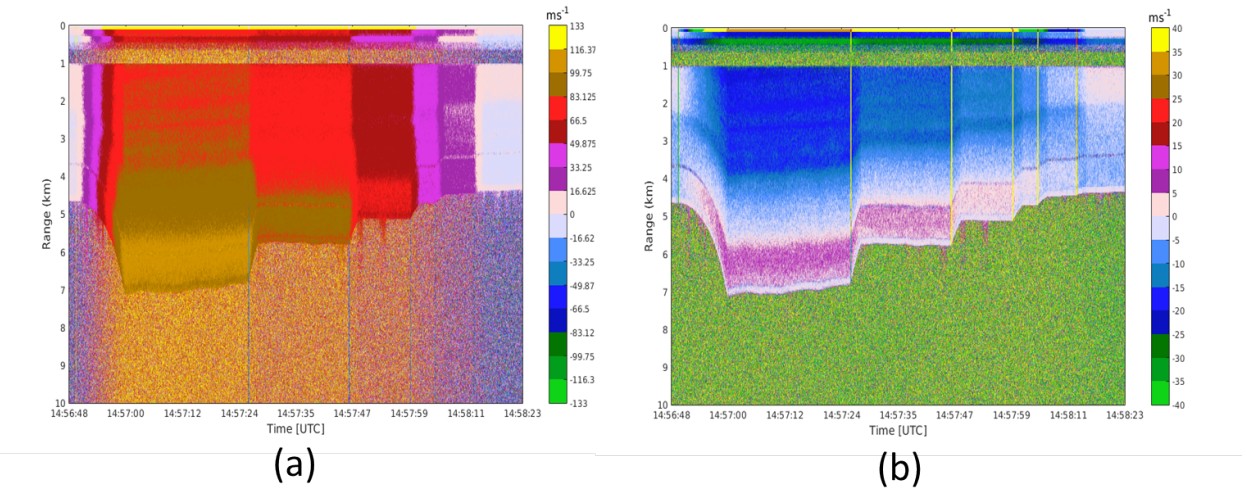

**Figure 16.** Velocity fields for the case study in Fig. 15: (a) measured PDPP velocity and (b) estimated velocity of the precipitation along the direction of the antenna beam.





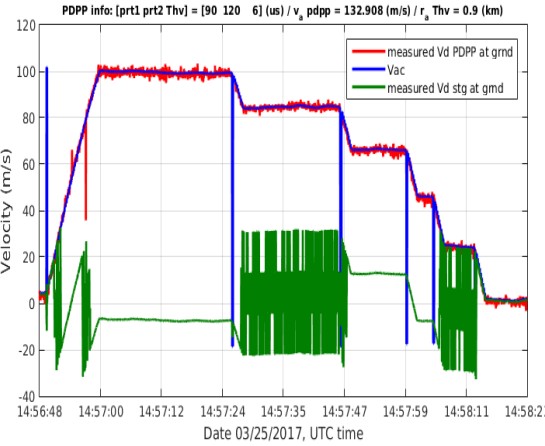

**Figure 17.** PDPP measured velocity (red line), the aircraft contribution computed from the aircraft INS data (blue line) and velocity estimates using staggered PRT pulses (green) at ground gates for the study case in Fig. 13.

## 5   Conclusions

This work describes the implementation of polarization diversity on the NRC Airborne W-band radar and the novel results collected during different flights conducted over North America in 2016 and 2017. This was the first time a PDPP mode has been implemented on a W-band radar on a moving platform. The conclusions of this study can be summarized as follows:

- A comprehensive PDPP I&Q dataset has been collected, which allows characterizing PDPP-based Doppler velocity estimates in various environmental conditions.

- The polarization diversity technique allowed much larger velocity to be measured unambiguously. Doppler velocities exceeding 100 m/s were measured during the field campaign when adopting a pulse pair separation $T_{hv}$ equal 6 $\mu$s.

- Cross-talk between the two polarizations caused by depolarization at backscattering deteriorated the quality of the ob-
10
  servations by introducing "ghost echoes" in the power signals and by increasing the noise level in the Doppler measurements. The regions affected by cross-talk were generally associated with the strongly depolarizing surface returns and to the depolarization of hydrometeors located at short ranges from the aircraft.

- The increased variance in Doppler velocity estimates were well predicted in cases where the signal to noise and signal to ghost ratios are known [Eq. (6-9)] or in cases where the observed correlation between the H-polarized and V-polarized
15
  successive pulses was measured [formula (11)]. The first approach can be used in simulation frameworks and end-to-end simulators; the second can be used to estimate the quality of the Doppler measurements directly from the observations themselves.





- The airborne field campaign has also provided novel observations of the backscattering properties of sea and land surfaces at W-band at viewing angles larger than 30°. This is the topic of a companion paper (Battaglia et al., 2017).

The measurement of 3D atmospheric winds in the troposphere remains one of the great priorities of the next decade (The Decadal Survey, 2017) and polarization diversity offers a solution to the short decorrelation expected from fast moving plat-

5   forms. Different concepts are currently being examined by different agencies (Durden et al., 2016; Illingworth et al., 2018a). This study provided a full proof-of-concept for an airborne W-band Doppler radar equipped with polarization diversity and therefore represents a key milestone towards the implementation of polarization diversity in space.

*Acknowledgements.* This work was supported in part by the European Space Agency under the activity Doppler Wind Radar Demonstrator (ESA-ESTEC) under Contract 4000114108/15/NL/MP and in part by CEOI-UKSA under Contract RP10G0327E13. We acknowledge the

10   support from the NRC flight and support staff during the flight campaign. We would like also to thank the Carleton University Co-op student Kenny Bala, who generated some of the figures used in the paper.





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
