# Peer review of "Implementation of Polarization Diversity Pulse Pair Technique using airborne W-band radar"

_Atmospheric Measurement Techniques, 2018_

## Referee Comment (RC1) · Anonymous Referee #2 · 3 Oct 2018

This paper describes the implementation and evaluation results of Polarization Diversity Pulse Pair (PDPP) technique on a W-band (94 GHz) airborne radar. In Earth Science research, atmospheric wind and cloud/precipitation Doppler velocity measurements using spaceborne Doppler radar are still challenging due to the Doppler spectrum broaden caused by platform ground speed and turbulence in the weather target system. PDPP has been proposed for future spaceborne radar to mitigate the challenge in Doppler measurements. Although PDPP has been used for ground-based radar, its performance on a fast-moving platform, such as aircraft and spacecraft is still needed to be studied. This paper discusses the benefits and limitations of PDPP approach for spaceborne application based on airborne measurements. In my view, it represents a substantial contribution to research progress that is in line with the scope of this journal. This paper is well written and organized. The technical approach is valid and results are in good quality. Therefore, I would recommend for publication after revision.

1. EarthCARE CPR is designed for cloud vertical velocity measurements while WIVERN is target for horizontal wind retrieval. Have the author done or planned to do any horizontal wind retrieval based on airborne data?
2. Figure 17, why the surface velocity estimated by staggered PRT technique (green curve) folded at some antenna position, but not the others.?
3. What is the cross-pol isolation of NAWX channels? The SNR in the data case shown in Figure 9 seems high so the surface contamination from cross-pol seems not very significant. Is there any data case with weaker cloud/precipitation layer(s)?

4. Page 2, Line 79, "94Ghz" to "94GHz".
5. Page2, Line 105, "… a low pair repetition frequency (PRF)…" to "… a low pulse repetition frequency (PRF) …".
6. Page2, Line 129, please spell the full name of "WIVERN".
7. Page 3, NAW, NAWX (and later NAX) are used through out of this paper. Please clarify that NAW is W-band alone, NAX is X-band alone in case of confusion.
8. Page 3, Line 189, "(Zrnic and Mahapatra, 1985) have …" to "Zrnic and Mahapatra (1985) have …".
9. Page 3, Line 204, "… ; (assuming a …" to "… ; assuming a …".
10. Page 3, Line 207, "… greater than 0.9 …", what greater than 0.9? $\rho$ ?
11. Page 5, Table 2, "[-40º-40º]" to , [-40º to 40º]" .
12. Page 5, Table 3, the $T_2/T_3$ ratio is 3:4, 5:6. But on Page 3, Line 190-194, it states that $T_2/T_3$ = 2:3 is optimal and the PDPP waveform was also designed for 2:3. Need to be consistent.
13. Figure 4, Has the water vapor and gas attenuation been corrected?
14. Figure 5, "The NRC Convair …" to " (a) The NRC Convair …". "… show aircraft heading, roll angle, beam incidence angle and aircraft altitude." to "… show aircraft (b) pitch, (c) roll angle, (d) beam incidence angle and aircraft altitude."
15. Page 8, Line 316, "Wivern" to "WIVERN".
16. Page 8, Line 340-344, PVV and PHH could be different for large rain drops and for surface return at high incidence angle.
17. Page 10, Line 438, what is "PP10 mode"?

18. Page 10, Line 450, 454 (and later in the paper), at X-band dBZ is used instead of $Z_e$.
19. Page 11, Figure 9 caption, "… an V-pulse…" to " … a V-pulse…".
20. Page 12, Line 505, "… (see (Torres et al., 2004) for details)." to "… (see Torres et al., 2004 for details)."
21. Page 12, Line 511, "Wivern" to "WIVERN".
22. Page 12, Line 512, and Page 14, Line 619, "$V_d$" to " $v_D$" so it will be consistent with the equations.
23. Page 12, Line 565, delete the sentence "Again it is …" (duplication of Line 561).
24. Page 12, Line 600, "Fig. 13 shows PDPP…" to "Fig. 15 shows PDPP…"
25. Page 15, Figure 15, the color table is highly compressed from range -20 dBZe to 68 dBZe. Could use -20 dBZe to 40 dBZe?. Also please change "dBZ" to "dBZe" for W-band reflectivity.
26. Page 15, Figure 16 caption, "(b) estimated velocity of the precipitation along the direction of the antenna beam" to "(b) estimated velocity of the precipitation along the direction of the antenna beam based on PDPP measurements and after removal of aircraft motion".

---

## Referee Comment (RC2) · Anonymous Referee #3 · 30 Oct 2018

This paper describes the first application of the polarization diversity pulse pair processing technique to derive Doppler measurements from the air. This is a first step towards the application of the technique to measure 3D winds from a space based platform. While the paper highlights some deficiences, the advantage of the technique is in the measurement of unambiguous velocities exceeding 100 m/s. The paper is well written and my comments are entirely technical.

As a general comment, "antennas" is typically used as plural in reference to radars, "antennae" in relation to bugs. Both are technically correct and authors choice.

Figure 1 - lists top and bottom panels, where they are actually laid out side by side

Figure 2 - caption does not refer to to (a), (b), (c) and (d)

[Figure]

Page 5 line 2 - add "the" in front of radar

Page 6 line 1 - add "as" to read "such as at the antennae"

Page 6 line 8 - "clear air condition" should have an s on the end

Page 6 - Figure 4 is referred to before Figure 3, and similarly Table 3 before Table 2. Consider swapping the figures and tables so they are referred to in order.

Figure 4 - the date at the top of panel a says 2016, while the caption says 2017, and panel b has no date

Figure 5 - refer to caption letters

Page 13 line 1 - I think should read modular rather than module?

Page 13 line 5 - remove capitalisation from A CloudSat

Page 14 - equation 9 also has a reference to equation 8

Page 15 line 7 - extra/missing bracket

Page 16 line 1 - has PP10 mode been defined?

Page 16 line 2 - add "the" to read "for the remainder of the flight duration"

Page 16 line 17 - add a full stop after aircraft

Page 19 line 11 - add "a" to read "there are a few aliased points"

Page 19 line 32 - change "allows to estimate" to "allows an estimate" or similar

Page 21 line 5 - add a space between limit and the bracket

page 21 line 7 - add a space before the bracket

Page 21 line 9 - add "the" to read "can be seen at the beginning"

Page 22 line 3 - add an s to level

---

## Author Comment (AC1) · 19 Nov 2018

We would like to thank the reviewer for these very helpful and constructive comments. Please find our detailed responses to your comments as follows.

1.   *EarthCARE CPR is designed for cloud vertical velocity measurements while WIVERN is target for horizontal wind retrieval.  Have the author done or planned to do any horizontal wind retrieval based on airborne data?*

The objective of the project was to demonstrate the PDPP technique from a moving platform so our flight profiles were not optimized to do comparison of PDPP derived horizontal winds with vertical profile of horizontal winds. Even for the WIVERN satellite the scope is to measure horizontal winds along the line of sight and the information

being assimilated by NW models.

However, we did a limited qualitative comparison of the PDPP Doppler velocity with a horizontal wind component obtained from radiosonde at proximity of the flight and the in-situ wind data measured by the aircraft wind system. In Fig. 1 (see attached), we plotted the 12 Z soundings (approximately 2 hours before the PDPP data collection) of the horizontal wind speed and direction obtained from two nearby radiosonde stations. In spite of the time and location differences of the soundings, both the radiosonde and aircraft sounding show a similar horizontal wind profile as the region was influenced by large scale synoptic system. The winds between 2 and 5 km were WNW with speed decreasing with altitude from 25 m/s at the aircraft altitude to nearly 5 m/s at 2 km altitude. Below 2 km, the wind veers with decreasing altitude and becoming light ($<$ 10 m/s) north easterly. In this lower layer, the contributions of the terminal velocity from rain drops cannot be neglected so the Doppler velocity is a combination wind and particle fall velocity. These wind profiles are consistent with the PDPP observations showing a gradual decrease in magnitude from -20 m/s near flight level due to strong westerly winds (aircraft flying towards east) then reverse direction to weak vD of 0-10 m/s in a layer where with winds were north easterly (opposite to aircraft heading).

Figure 1(full caption): Left: vertical profiles of horizontal wind speed obtained during the Convair descent towards Ottawa after the PDPP data collection (black), 12Z upper air sounding from Buffalo (US) and Maniwaki (Canada) and profile of PDPP VD after aircraft motion correction (blue). Right: wind direction corresponding to the aircraft and BUF and WMW soundings. The vertical line correspond to the aircraft heading and the horizontal like is perpendicular to aircraft heading to the WVM sounding.

2. *Figure 17, why the surface velocity estimated by staggered PRT technique (green curve) folded at some antenna position, but not the others?*

The maximum Doppler velocity for the staggered PRT waveform in this case was 26.6 m/s. When the aft antenna was steered to a large slant angle, the Doppler velocity

introduced by the aircraft motion increased (up to over 100 m/s). When the signal radial Doppler (aircraft contribution plus precipitation Doppler) exceeded 26.6 m/s, staggered PRT velocity started folding.

3. *What is the cross-pol isolation of NAWX channels? The SNR in the data case shown in Figure 9 seems high so the surface contamination from cross-pol seems not very significant. Is there any data case with weaker cloud/precipitation layer(s)?*

The cross-polarization isolation of the W-band channels mainly limited by the cross-pol of the antennas which is 30 dB for E-plane and 36 dB for H-plane. We have collected a few data sets where the SNR of cloud layers was low and cross-pol signals were much more significant (see Figs. 2-4). In those cases, Doppler velocities showed large errors or were not available in regions contaminated by cross-pol signals.

Figures 2, 3 and 4: PDPP $Z_{hh}$, $Z_{hv}$ and retrieval Doppler for the March 29th, 2016 case.

Comment 4-26: We thank the reviewer for pointing these out. The errors/typos have been corrected.

4. *Page 2, Line 79, "94Ghz" to "94GHz".* - corrected as suggested

5. *Page2, Line 105, ". . . a low pair repetition frequency (PRF). . ." to ". . . a low pulse repetition frequency (PRF) . . .".* - corrected as suggested

6. *Page2, Line 129, please spell the full name of "WIVERN".* - WIVERN is spelled out as suggested

7. *Page 3, NAW, NAWX (and later NAX) are used through out of this paper. Please clarify that NAW is W-band alone, NAX is X-band alone in case of confusion.* - We have defined NAX and NAW

8. *Page 3, Line 189, "(Zrnic and Mahapatra, 1985) have . . ." to "Zrnic and Mahapatra (1985) have . . .".* - corrected as suggested

9. *Page 3, Line 204, "... ; (assuming a ..." to "... ; assuming a ...".* - corrected as suggested

10. *Page 3, Line 207, "... greater than 0.9 ...", what greater than 0.9?* - $\rho$ is defined as normalized signal correlation in Eq. 1, so $\rho > 0.9$ means normalized signal correlation greater than 0.9

11. *Page 5, Table 2, "[-40º -40º]" to , [-40º to 40º]"* - corrected as suggested

12. *Page 5, Table 3, the $T_2/T_3$ ratio is 3:4, 5:6. But on Page 3, Line 190-194, it states that $T_2/T_3$ = 2:3 is optimal and the PDPP waveform was also designed for 2:3. Need to be consistent.* - We thank the reviewer for pointing this out. We added text to the revision to make this point clear: "Therefore, the PDPP waveform was designed such that $T_2/T_3$ is close to 2/3. Additionally, the pulse spacing $T_2$ and $T_3$ were set according to the maximum desired measurement range - velocity and the transmitter duty cycle limit of radar."

13. *Figure 4, Has the water vapor and gas attenuation been corrected?* - Gas attenuation and water vapour corrections have not been applied, since the flight in question was performed at a sufficiently low altitude (about 1 km) in clear air condition . As a result, the effects of water vapour and gas attenuation are minimal.

14. *Figure 5, "The NRC Convair ..." to " (a) The NRC Convair ...". "... show aircraft heading, roll angle, beam incidence angle and aircraft altitude." to "... show aircraft (b) pitch, (c) roll angle, (d) beam incidence angle and aircraft altitude."* - corrected as suggested

15. *Page 8, Line 316, "Wivern" to "WIVERN".* - corrected as suggested

16. *Page 8, Line 340-344, PVV and PHH could be different for large rain drops and for surface return at high incidence angle.* - We completely agree with the reviewer. That assumption was used for a calibration case, and does not need for the analysis in this paper. We removed the equation and revised the text accordingly.

17. *Page 10, Line 438, what is "PP10 mode"?* - We added a text in the caption of Fig. 1 that described the NAW PP10 mode.

18. *Page 10, Line 450, 454 (and later in the paper), at X-band dBZ is used instead of $Z_e$.* - We modified the text for better presentation "Reflectivity from the up and down antennas of the NRC X-band airborne radar . . ."

19. *Page 11, Figure 9 caption, ". . . an V-pulse. . ." to " . . . a V-pulse. . .".* - corrected as suggested

20. *Page 12, Line 505, ". . . (see (Torres et al., 2004) for details)." to ". . . (see Torres et al., 2004 for details)."* - corrected as suggested

21. *Page 12, Line 511, "Wivern" to "WIVERN".* - corrected as suggested

22. *Page 12, Line 512, and Page 14, Line 619, "$V_d$" to "$v_D$" so it will be consistent with the equations.* - corrected as suggested

23. *Page 12, Line 565, delete the sentence "Again it is . . ." (duplication of Line 561).* - corrected as suggested

24. *Page 12, Line 600, "Fig. 13 shows PDPP. . ." to "Fig. 15 shows PDPP. . ."* - corrected as suggested

25. *Page 15, Figure 15, the color table is highly compressed from range -20 dBZe to 68 dBZe. Could use -20 dBZe to 40 dBZe?. Also please change "dBZ" to "dBZe" for W-band reflectivity.* - We updated the color scale as suggested.

26. *Page 15, Figure 16 caption, "(b) estimated velocity of the precipitation along the direction of the antenna beam" to "(b) estimated velocity of the precipitation along the direction of the antenna beam based on PDPP measurements and after removal of aircraft motion".* - corrected as suggested

[Figure]

**Fig. 1.** 12 Z soundings the horizontal wind speed and direction obtained from two nearby radiosonde stations

[Figure]

**Fig. 2.** PDPP Z_hh for the March 29th, 2016 case

[Figure]

**Fig. 3.** PDPP Z_hv for the March 29th, 2016 case

measured PDPP Doppler velocity
PDPP info: [prt1 prt2 Thv] = [90  120    6] (us) / v_a pdpp = 132.9222 (m/s) / r_a Thv = 0.9 (km)
lat: 48.6233 ; lon: -68.1927 ; mirror angle: 29.9866

**Fig. 4.** PDPP retrieval Doppler for the March 29th, 2016 case

---

## Author Comment (AC2) · 19 Nov 2018

We would like to thank the reviewer for the comments and we have accepted most of the suggestions. Please find our detailed responses to your comments as follows.

1)  *As a general comment, "antennas" is typically used as plural in reference to radars, "antennae" in relation to bugs.  Both are technically correct and authors choice.* - changed as suggested

2)  *Figure 1 - lists top and bottom panels, where they are actually laid out side by side* - Thank you for pointing that out. When the paper was formatted from double column to single column the figures were rearranged. But now we corrected the label.

3)  *Figure 2 - caption does not refer to to (a), (b), (c) and (d)* - We added a text corre-

sponding to the figure letters

4) *Page 5 line 2 - add "the" in front of radar* - changed as suggested

5) *Page 6 line 1 - add "as" to read "such as at the antennae"* - changed as suggested

6) *Page 6 line 8 - "clear air condition" should have an s on the end* - changed as suggested

7) *Page 6 - Figure 4 is referred to before Figure 3, and similarly Table 3 before Table 2. Consider swapping the figures and tables so they are referred to in order.* - The order of figures changed as suggested. The order of Tables, however, had been correct as the Table 2 is first referred on page 4

8) *Figure 4 - the date at the top of panel a says 2016, while the caption says 2017, and panel b has no date* - The typo in the caption was fixed, and the date from the top of the panel removed. Please note that two flights, on March 29, 2016 and on March 4, 2017 are discussed.

9) *Figure 5 - refer to caption letters* - changed as suggested

10) *Page 13 line 1 - I think should read modular rather than module?* - changed as suggested

11) *Page 13 line 5 - remove capitalisation from A CloudSat* - changed as suggested

12) *Page 14 - equation 9 also has a reference to equation 8* - unnecessary number was removed

13) *Page 15 line 7 - extra/missing bracket* - changed as suggested

14) *Page 16 line 1 - has PP10 mode been defined?* - The definition was added in the caption of Fig.1

15) *Page 16 line 2 - add "the" to read "for the remainder of the flight duration"* - changed as suggested
16) *Page 16 line 17 - add a full stop after aircraft* - changed as suggested

17) *Page 19 line 11 - add "a" to read "there are a few aliased points"* - changed as suggested

18) *Page 19 line 32 - change "allows to estimate" to "allows an estimate" or similar* - changed as suggested

19) *Page 21 line 5 - add a space between limit and the bracket* - changed as suggested

20) *Page 21 line 7 - add a space before the bracket* - changed as suggested

21) *Page 21 line 9 - add "the" to read "can be seen at the beginning"* - changed as suggested

22) *Page 22 line 3 - add an s to level* - changed as suggested